# CoR-SketchAR: Cooperative Sketch-Based Real-Time Augmented Reality Authoring Tool for Crowd Simulation

**Gahyeon Kim and Mankyu Sung \***

Department of Faculty of Computer Engineering, University of Keimyung, Daegu 42601, Korea; gh87913711@gmail.com
* Correspondence: mksung@kmu.ac.kr

**Abstract:** In this study, we propose CoR-SketchAR, an augmented reality (AR) environment authoring tool that uses dry-erase markers for real-time collaboration among users. The most important requirement for multi-user collaboration in an AR environment is that the 3D virtual object must be placed at a specific location and can be easily changed by multiple users. Because marker-based registration techniques, which are widely used for matching virtual objects with real ones, require a marker for each object, and creating a crowd simulation environment with objects of various shapes and sizes requires the use of a large number of markers, which is time consuming and expensive. CoR-SketchAR, instead, creates an urban AR environment by drawing sketches with easily altered dry-erase marker. Then, system recognizes the shapes and colors of the sketches automatically. Those recognized shapes and colors provide the exact positions for overlaying the 3D virtual objects, which are the environment factors, on the real environment in augmented reality manner. We can even specify the path the crowd are moving along with a simple sketch stroke. We apply the computer vision technique to recognize the colors and shapes of sketches. By altering the size, shape and color sketches, the system is able to create a wide variety of dynamic urban environments. To validate the proposed techniques, we built two stand-alone software systems to check the usability of the proposed system (a 2D screen-based environmental authoring tool and a sketch-based environmental authoring tool) and conducted experiments in which two users collaborated with each other to create an environment with a specific authoring tool and then report surveys. In the experiments, users collaborated in pairs to create environmental elements, such as highways, buildings, trees, and the starting and goal positions of crowds. After recognizing them, the system then automatically creates a 3D environment, and crowds are animated accordingly. Based on a user survey, we observed that participants who used sketch-based environmental authoring tools were more active and accessible than those who used 2D screen-based authoring tools. The results of the study show that CoR-SketchAR can be further used to create a dynamical crowd simulation on a large scale using beam projectors or portable devices by simply adding sketches based on different scenarios.

**Keywords:** augmented reality; crowd simulation; sketch-based authoring tools





## 1. Introduction

To reduce scene-making costs, virtual crowd simulation has been widely used in the entertainment industry, such as in films and games. In this simulation, because crowd movement is constrained by the surrounding environment, it is important to create an environment in which a crowd behaves and reacts accordingly. Factors affecting the environment must first be considered to create an environment for crowds.

By definition, the elements that constitute the environment are referred to as the environmental factors. This is owing to the fact that environmental factors have a significant impact on crowd behavior. For example, people line up to take a bus in front of a bus stop. They wait for signals to cross at a crosswalk and sometimes find a gate to enter and leave the building once they finish their business there. Thus, specific environmental factors

can restrict people's behavior and create a new mode of crowd behavior. For instance, buildings, trees, and roads must be constructed to control various crowd behaviors.

When we build environmental factors, their various shapes must be supported to create a large-scale space, such as the city center, which is expensive and time consuming. Furthermore, it is difficult to renovate or rebuild these structures. In general, miniatures, which are small models of the actual environment, have been built, placed, and reviewed in advance for city design. Miniatures are usually made and placed manually. Therefore, it is difficult to modify them once they have been created. Unlike physical miniatures, 3D virtual models can be created on computers, which has an advantage over miniatures because they allow users to create and modify objects by mouse manipulation. However, poor immersion is one of its disadvantages, because only a flat 2D screen can be viewed during modeling. To compensate for the lack of immersion, we propose an augmented reality (AR) technique that allows viewing of the actual model based on the user's location. In AR, image markers are used to overlap live-action objects with computer graphic (CG) objects. An image marker estimates the position of a real-life object such that it can be precisely overlaid with the CG object. Once registration is completed, it can be observed from various camera angles. In our system, we used an easy-to-carry portable mobile device to visualize the AR scene because it is easy to rotate the viewpoint and view direction. Using several appropriate 3D building models, we can create a small city environment. On top of the city environment, virtual crowds must be placed so that they can interact with the environment similar to the real world. In this study, a technique for simulating a virtual crowd based on sketches is proposed to provide a feeling of liveliness in urban environment design.

A critical problem of image-based markers in AR is that each marker can be applied to only a single virtual object. Therefore, to place many objects, the user will require to produce as many markers as the number of objects. In addition, pre-made markers are difficult to modify and have a fixed shape, making them unsuitable for constructing urban environments. To address these issues, we propose a CoR-SketchAR system that can add or remove a wide variety of user-drawn sketches representing the specific virtual objects as erasable marker pens on the whiteboard. The sketch drawing method is more convenient than the existing computer-based GUI method because pen and markers are more natural for human users.

Figure 1 shows the CoR-SketchAR system. In this example, a user creates a crowd simulation environment by drawing sketches with dry-erase markers of different colors and shapes. The sketches were then converted into 3D buildings, highways, trees, and buildings with exits by recognizing their shapes and colors. Not only those 3D objects mentioned above but sketch-based paths can also be specified using a black dry-erase marker to control the movement of the crowd. The 3D human character models were then placed at the location of the crowd along the path. During animation, collisions between characters are prevented automatically. Experiments confirmed that the proposed system had a higher level of participation and accessibility than general 2D screen-based environmental authoring tools. Furthermore, the survey revealed that CoR-SketchAR was more useful and easier to learn than 2D-authoring tools. The higher accessibility and participation of the system is, the higher the user's sense of immersion in the environment they are creating, which also can increase the efficiency of completing the given tasks.

The remainder of this paper is organized as follows. In Section 2, we introduce prior research approaches for applying AR and previous studies on augmented reality, including crowd simulation. In Section 3, we present detailed algorithms for shape recognition, environmental authoring tools, routing, and crowd generation. In Section 4, we perform a series of experiments for generating environments with the authoring tools we propose, and we conduct a comparative survey between the sketch-based AR environmental authoring tool and the computer-based environmental authoring tool. Section 5 examines the proposed method and discusses the future research directions.

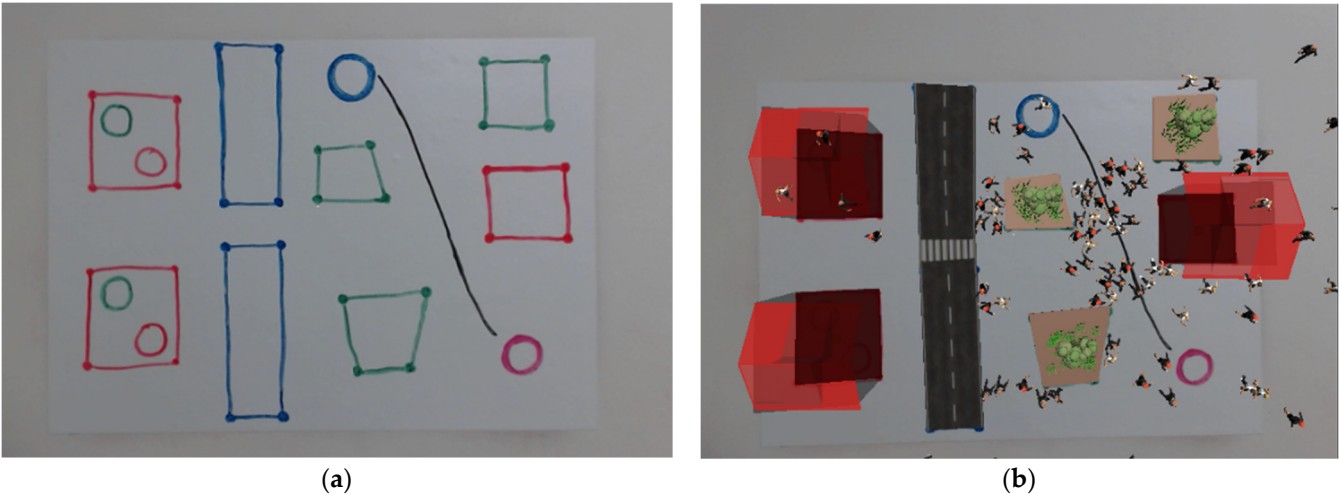

(**a**)              (**b**)

**Figure 1.** (**a**) Hand-drawn sketches of the users; (**b**) Project-based visualization of virtual environment where crowds are behaving based on the different environmental factors.

## 2. Related Work

### 2.1. Real-Time Cooperative System

Collaborated work increases work efficiency by sharing ideas and dividing works for users. Advances in information technology (IT) have raised the need for efficient collaboration work in IT systems. Various collaborative systems for IT technology are being developed to meet these requirements [1]. Platforms such as Google Drive, Figurema, Envision and Discovery Collaborator (EDC) of the University of Colorado and Github provide an environment where users can use software tools for collaboration between co-workers [2,3].

Among IT fields requiring collaboration, the development of web and smartphone modules that share 3D models enabled avatar-based video conferencing for multiple users and made decision making easy, which can reduce the repetitive discussion of conceptual design phases in 3D model manufacturing, resulting in faster design times [4]. Furthermore, to simplify the repetitive design process in the field of 3D modeling or urban environment design, a collaborative system based on AR was developed [5,6]. Another benchmark real-time design study was proposed by Seichter et al. [7]. In this work, when one user is designing the city, the others can only see the city through a 2D screen.

All methods mentioned above have a drawback in collaboration because it allows only one user to participate in the design of a job. To overcome these constraints, the CoR-SketchAR, on the contrary, applies useful dry-erase markers created by participants through sketches together in the design process. This makes the design of the urban environment easy to be shared, discussed, and changed.

### 2.2. Design Using Augmented Reality

Extensive research is being conducted to implement AR that makes virtual 3D models appear in real life. In general, AR uses cameras and recognizes specific markers to position virtual objects. When the camera detects the markers, it places the desired 3D model in the same location. An advantage of marker-based AR is that it allows users to view a virtual 3D model from multiple perspectives. In urban environment planning, for instance, ArUco markers have been applied to place various building 3D models in specific locations [8]. In addition, a marker similar to a floor plan was developed for users unfamiliar with complicated floor plans [9]. These systems were used to enable users to understand the building constructed using the floor plan. These AR systems can increase the immersive feeling for users. To make an even more life-like virtual space, researchers have begun to integrate the scenes with large crowds into the augmented reality space. Worst et al. used cube markers as obstacles to create a virtual crowd in a museum [10]. In this study, visitors

were able to place cube markers as architectural structures which acted as obstacles and affected crowd movement automatically. This allows visitors to become more immersed in constructing an urban environment through the interaction between the visitor and virtual environment. Subsequently, a beam projector was used to make the crowd visible to a large number of people. However, because it can only construct specific types of obstacles, the simulation is limited in that it cannot produce a variety of architectural structures.

To increase immersion of simulation, we also applied AR system in CoR-SketchAR. However, one significant advantage of our system over the marker-based AR system is its usability. Because the marker-based AR systems can place only a single 3D model per marker, it is impossible to place buildings, roads, and other structures in the city center with different shapes. For place various shapes of architectural structures in the city, we use instead a markerless-based method that recognizes objects based on undetermined information. Undetermined information is received through sketches which can easily change with dry-erase markers in the CoR-SketchAR system. These sketches are used as input data for AR visualization in our CoR-SketchAR system. Furthermore, the developed and modified results can be viewed using a beam projector and portable mobile in the form of AR, which can increase immersion.

### 2.3. Crowd Collision Avoidance

The key to crowd simulation is that no individual should bump into another during animations. Jur et al. proposed a technique for avoiding collisions between two moving objects by creating a reciprocal velocity obstacle (RVO) based on optimal reciprocal collision avoidance (ORCA) [11,12]. This algorithm avoids collisions by altering the speed and direction based on the ORCA region, which can calculate the best speed and direction for an object to avoid collision for given target points. The RVO technique has an advantage in bottleneck situation compared to other algorithms because it can predict the bottleneck beforehand and avoid it at earlier time. In addition, because the RVO technique makes potentially collided moving objects share the responsibility for speed and direction changes, it can reduce the resolving time when crowds are stuck. We applied RVO with the ORCA technique in this study to avoid crowd collisions and reduce the standstill time.

By extending this technique, Best et al. proposed a reciprocal velocity obstacle ellipse (RVOE) that represents individuals as ellipses instead of circles to approximate a real crowd more closely [13]. However, elliptical collisions have a disadvantage in terms of the computation time compared to circular collisions. Because CoR-SketchAR must run on portable devices with limited specifications, we used the RVO algorithm instead.

### 2.4. Path-Planning of Crowd

In crowd simulations, starting points, arrival points, and obstacles must be specified for the individuals. To create such an environment, it is common to use the simple graphic user interfaces (GUI), such as buttons or mouse clicking on the environment directly as shown in Figure 2. However, these solutions can be challenging for users unfamiliar with the computer environment. Stigall et al. proposed a crowd evacuation training simulation in AR [14]. In this study, they used GUI buttons on a portable device that controls the way the crowd leaves. The blue arrow button was also used to specify an emergency evacuation path. However, because the path can be specified with many arrows, the path between the arrows is sometimes invisible. These invisible paths are less intuitive for users. Our system, on the other hand, proposed a more intuitive sketch-based method, which can draw a curve using a black dry-erase marker to specify a crowd path. The path can be understood more intuitively.

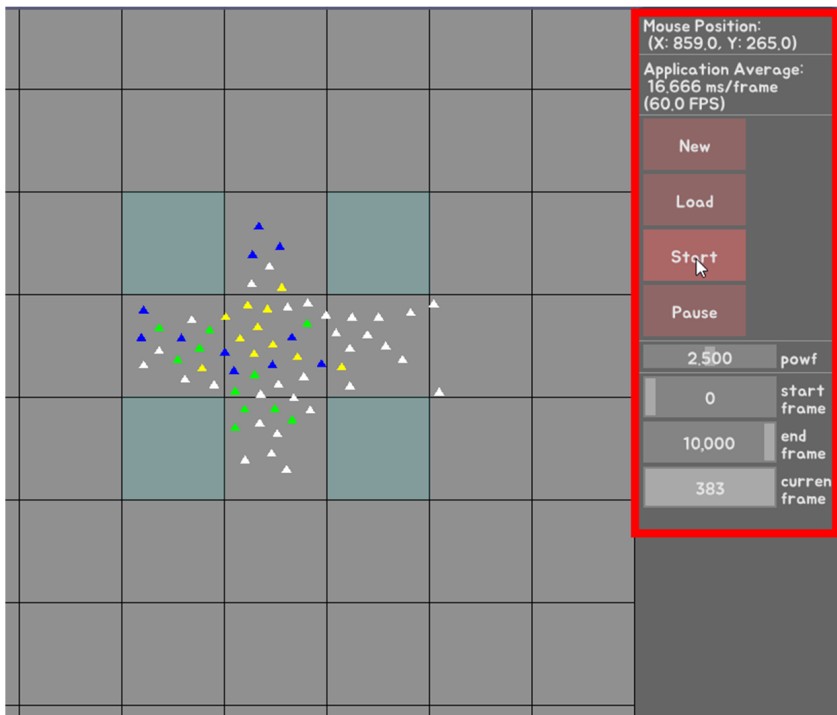

**Figure 2.** Graphic user interface (GUI) of crowd simulation in computer: All parameters for the simulation are set with widgets, such as sliders or buttons.

## 3. Algorithm

### 3.1. Overview

The hypothesis that this research suggests is that the use of sketch-based authoring tools has advantages in term of collaboration, visual quality and total time to complete over conventional 2D GUI-based authoring tools. In this chapter, we will prove the hypothesis by explaining the detailed algorithm and performing a set of experiments. The overall process of the CoR-SketchAR system is shown in Figure 3. First, the user sketches using whiteboards and erasable dry-erase markers. Then, the drawn sketches are classified into square and circular sketches using computer vision technology. Subsequently, the location information of the rectangular sketch is extracted from the classified information. A virtual environment is then created by placing a 3D object on each rectangular stroke. Thereafter, circular strokes that indicate the initial and goal positions of the crowds are set. Additionally, a black curve stroke can be drawn, which represents the path along which the crowd should move between the initial and goal positions. Finally, the visualization module displays the 3D building environment and the 3D crowd animation.

### 3.2. Sketch Recognition for Environment Setting

To translate sketches into a crowd environment, users have to sketch shapes with erasable dry erase markers on a whiteboard. Different shapes and colors of user-drawn strokes provide different environmental factors, such as roads, trees, and buildings.

To recognize the particular shape of a stroke, we must first consider the number of vertices of the shape. Vertex count was estimated using the Douglas–Peucker algorithm [15]. Once the camera captures a sketch, lots of vertices may be detected. The Douglas–Peucker algorithm allows us to classify a linear line from a set of detected points by approximating the distance between vertices. Based on the searched vertex count, we find a shape with four vertices among all approximated vertices. Because a circle consists of a lot of vertices, recognizing the vertex count takes a lot of computing time. Therefore, in our approach, circles are recognized through the Hough gradient method [16]. This method is based on Hough transformation but also measures the slope at the edge to determine if the point is relevant to the circle or not [16]. Subsequently, the colors of the sketched shapes are

recognized. To minimize the error caused by different light conditions, we transformed the RGB color space into the HSV space based on [17].

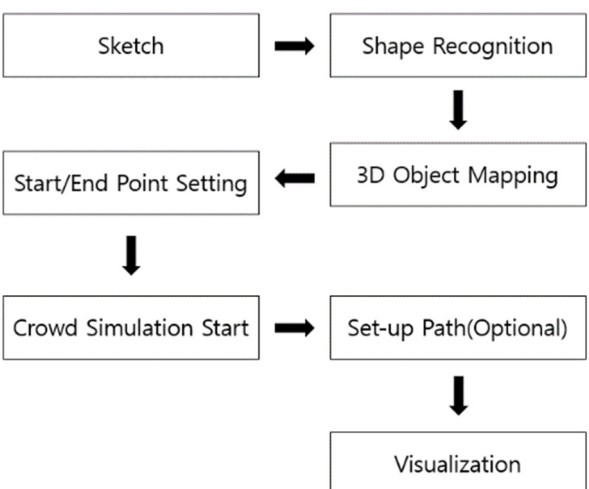

**Figure 3.** Overview of the CoR-SketchAR system.

Because the brightness and color information of an image can be separated in the HSV space, the color information can be used only in the recognition process because brightness affects the color value depending on the light condition. Note that H in the HSV color space is color, S is saturation, and V is brightness. The HSV value ranges are shown with the hue values in Table 1.

**Table 1.** Color ranges of webcam and mobile device.

| Color Range of Webcam | | | |
|---|---|---|---|
| **Color** | **Hue** | **Saturation** | **Value** |
| Red | 171–180 | upper 30 | upper 130 |
| Green | 80–100 | upper 30 | upper 130 |
| Blue | 101–120 | upper 30 | upper 130 |
| Pink | 140–170 | upper 30 | upper 130 |
| Black | 0–180 | under 20 | under 130 |
| **Color Range of Mobile Device** | | | |
| **Color** | **Hue** | **Saturation** | **Value** |
| Red | 0–60 | upper 20 | upper 200 |
| Green | 61–80 | upper 20 | upper 200 |
| Blue | 81–180 | upper 20 | upper 200 |
| Pink | 160–169 | upper 20 | upper 200 |
| Black | 0–180 | under 20 | under 100 |

Through shape and color recognition, the starting point was recognized as a blue circle, and the destination point was recognized as a pink circle. Roads, trees, and buildings were recognized as blue, green, and red quadrilaterals, respectively. The crowd paths are represented by the black curves.

Among all the environmental factors, buildings in the city are divided into two categories. One type comprises buildings where crowds can enter or exit, and the other type comprises buildings that people cannot enter. People typically enter a building through an exit and then exit the building through a different exit. Therefore, a building with an exit was set as an accessible space. An accessible building was created by sketching a green and red circle inside the red quadrilaterals, recognized as walls of the building. The overall shapes and color information for the recognition of the sketches are listed in Table 2.

**Table 2.** Categorization of objects by shape and color.

| Object | Shape | Color |
|---|---|---|
| Start Point | | Blue |
| Goal Point | | Pink |
| Exit Up | | Green |
| Exit Down | Circle | Red |
| Road (for car) | | Blue |
| Tree | | Green |
| Building | Quadrilateral | Red |
| Path | Line | Black |

### 3.3. D Object Mapping

The CoR-SketchAR system performs position mapping such that the recognized shapes can be replaced by 3D objects at runtime. When we place 3D objects at the center place of the recognized sketch points, we must obtain four vertices of the shapes, denoted as vertices 0–3, as shown in Figure 4. The upper vertex numbers 4–7 change depending on the height of the 3D object. Subsequently, additional 3D parts, such as roofs or trees, are placed above them.

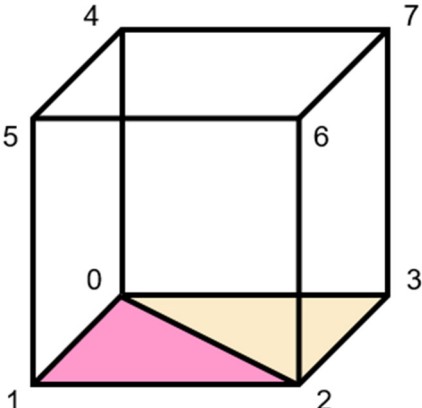

**Figure 4.** Vertex order of the 3D object: the number indicates the order of vertices to build 3D object. Once the positions of vertex number 0–3 are determined, then the position of vertex 4–7 are computed automatically from the height parameter.

Buildings with exits are created differently. They are installed using four rectangular walls on four sides as shown in Figure 5, and the method for installing the wall uses the following set of points from (1)–(4). The 2D points in the curly brackets represent the four corner vertices of the wall. In this set, $O_L$, $O_R$, $O_T$, and $O_B$ are the four surrounding walls, where $O_L$ is the left wall of the building, $O_R$ is the right wall, $O_T$ is the upside wall, and $O_B$ is the bottom wall. $EU$ is the center point of the green circle, representing the entrance, whereas ED is the center point of the red circle, which is the exit. $e$ is the thickness of the building wall.

$$O_L = \{\{V0_x, V0_y\}, \{EU_x + e, V0_y\}, \{EU_x + e, V3_y\}, \{V3_x, V3_y\}\} \tag{1}$$

$$O_R = \{\{V2_x, V2_y\}, \{V1_x, V1_y\}, \{V1_x - e, V1_y\}, \{V2_x - e, V2_y\}\} \tag{2}$$

$$O_T = \{\{EU_x - e, V1_y\}, \{V1_x, V1_y\}, \{V1_x, V1_y - e\}, \{EU_x - e, V1_y - e\}\} \tag{3}$$

$$O_B = \{\{V3_x, ED_y + e\}, \{ED_x + e, ED_y + e\}, \{ED_x + e, ED_y - e\}, \{V3_x, V3_y\}\} \quad (4)$$

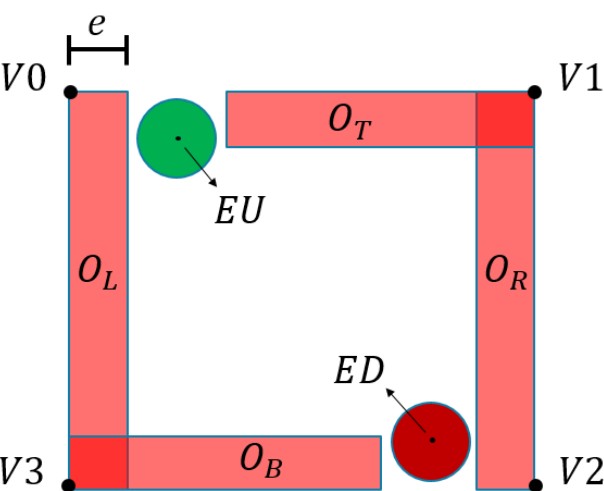

**Figure 5.** Details of building which have entrances (green circles) and exits (red circles).

### 3.4. Crowd Path Planning between Start and Goal

When users want to move a group of characters to a specific point, the path between the initial and goal points must be specified. One problem is that we need an automatic method to find separate collision-free paths for each individual to the goal positions. Our basic policy is that if a user specifies a rough path that crowds move along with a single sketch, then the algorithm automatically finds a collision-free path for each individual. Path planning is often used to simulate not only the evacuation situations of crowds but also the various behaviors of crowds.

The starting and goal points of the crowd are set using circular strokes, as listed in Table 2. Subsequently, a rough path is drawn as a black curve between the initial and goal positions by the user. If there is no user-specified path, the crowd moves in a straight line, which is the shortest distance from the starting point to goal point.

A problem that needs to be solved is the recognition of the hand-drawn path between the starting point and goal point. Because the black dots are recognized individually, we must sort them in the order of the starting point to ending point. In addition, among all the cluttered points, sampling is required to select evenly distributed points.

In our method, we assumed that the path consists of a set of black dots. Let us assume that we store all dot positions $p$ of the path into a set $P$. Among them, we perform a sampling so that we select a subset $S$ from $P$. The points in set $S'$ maintain a minimum distance from each other.

Mathematically, we can denote the point S as follows:

$$S = \{p | p_i \in P, |p_i - p_{i+1}| > t\}, \quad (5)$$

where $t$ represents the minimum distance from the neighbor points that each point maintains.

However, when we create point set $S$ as shown in Figure 6b, it is not sorted in the order along the path. Therefore, all points in the set must be sorted from the starting point to ending point. Figure 6c,d show a projection that aligns point set $S$ onto the vector from the starting to end points to obtain a new point set $S'$. Sorting is performed as follows. After the projection of the point set in $S$ is performed, the distance $D_i$, representing the distance from the starting point $St$ to point in $S'$, is calculated using Equation (6) as follows:

$$S_i' = ||S_i - St|| \cdot \cos(\theta), \quad (6)$$

where $\theta$ represents the angle between vector $S_i - St$ and vector p, $S_i$ represents the $i$-th point in point set $S$, and $St$ represents the starting point. Figure 7 shows this calculation.

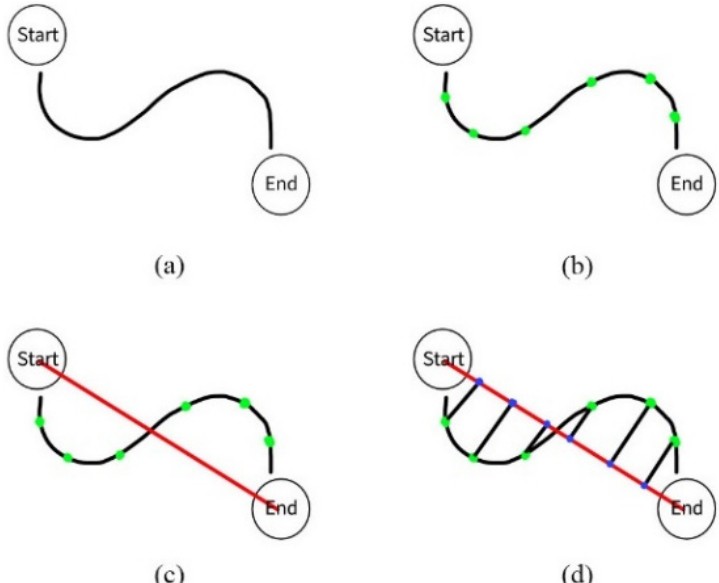

**Figure 6.** (**a**) A sketch showing the starting point, goal point, and path between them. (**b**) Green dots represent the midpoint of the path at regular interval. (**c**) A vector (red line) from the starting point to the goal point. (**d**) Projection of midpoints onto the vector in (**c**).

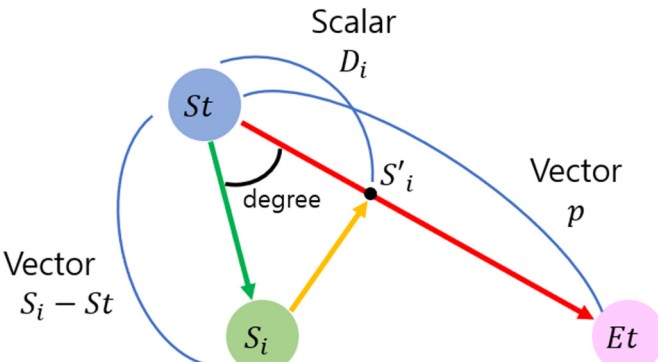

**Figure 7.** *i*-th point of the point ($S_i$) set *S* is projected onto the vector from *St* to *Et* denoted as a red line. The projected points ($S'_i$) are then sorted later.

### 3.5. Crowd Simulation

After setting the path between the starting and ending points, the crowd moves along the path during the simulation. During this process, interpersonal collisions must be prevented. In this study, we applied the RVO algorithm proposed by Jur et al. to avoid conflicts when individuals moved along a path [12].

The RVO technique adjusts the speed of multiple moving objects based on the velocity obstacle (VO), which is the region of the velocity domain in which all velocities of a moving object result in a collision with another moving object at a certain moment in time. Figure 8 illustrates the VO of object B for object A. The shaded area (a) represents the VO in the current frame of the simulation. The VO has a cone-shaped region where two tangents are obtained from point A to the disk whose center position is at point B, and the radius is the sum of the sizes of A and B. After the frame, the VO moves a step, as indicated by the velocity of object B, which is $V_B$, and the entire region is shown as the shaded area in (c). Therefore, when Object A selects the velocity $V_A$ that avoids the shaded area (c) in the next step, Object A is guaranteed to avoid Object B. If the next position of the neighboring moving object does not enter area (a), then it does not collide with the object completely. In this manner, each object moves toward its goal while avoiding collisions by adjusting its speed so that it does not invade each other's VO areas. When this scheme is

applied to a crowd, bottlenecks may occur when it is necessary to calculate the VO areas for several moving objects. To solve this bottleneck, Jur et al. proposed an RVO technique [12]. RVO improved the speed of crowd simulation by reducing bottlenecks by dividing the burden of adjustment speed by half with the relative object when determining the speed at which mutual collisions can be avoided. Please refer to Jur et al. for more details [12]. The CoR-SketchAR system produced collision-free crowd movement based on RVO while reducing the bottleneck.

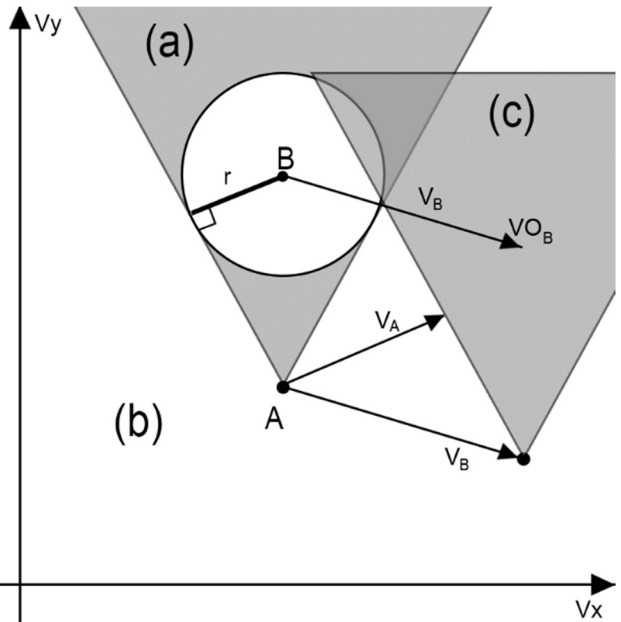

**Figure 8.** (**a**) Region of a velocity obstacle; (**b**) outside region of the velocity obstacle; (**c**) next frame of the velocity obstacle.

*3.6. Visualization*

The visualization step proceeds after completing all the environment settings using the CoR-SketchAR system. Visualization can be performed using two AR methods. The first method involves visualization using an AR beam projector. As shown in Figure 9, a beam projector was set behind the camera and a webcam was installed in front of it. The 2D screen data transmitted through the webcam were then processed on a computer and fed to the top view through the beam project.

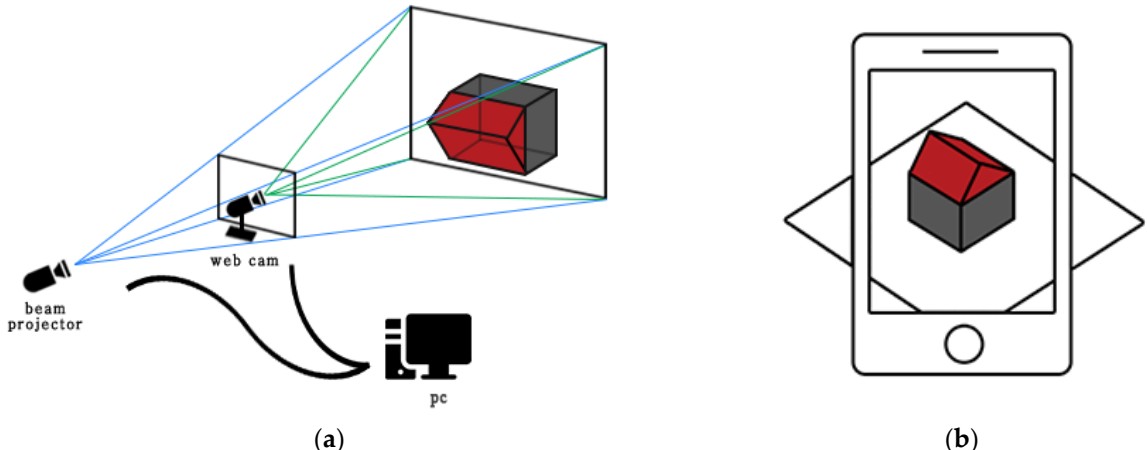

(**a**)            (**b**)

**Figure 9.** (**a**) Visualization with beam projector AR; (**b**) visualization with portable device AR. Note that portable device AR can change the viewpoint by moving the phone to other positions.

However, this visualization is limited. It cannot be observed from various angles because it provides only a top view. The second additional visualization AR method, the mobile AR visualization method, is designed to receive 2D screen data from the front camera of a mobile device and then view it from various angles.

## 4. Experimentation

We perform a series of experiments to verify the proposed system. We first explain the hardware setup, and then discuss the experimental results and surveys that participants turned in after the experiments.

### 4.1. Implementation

The hardware setup for the experiments included an Intel i7-9700K CPU, NVIDIA GTX 2070 GPU, and 16 GB RAM. The prototype system was developed on Windows 10, and free software libraries, such as Unity3D engines, OpenCV, and RVO, were used. For visualization, the Logitech C920 PROHD webcam, Optoma LED Projector LDMLUUZ, and portable mobile device Galaxy S9 were used to show the AR. Dry-erase markers were used on the whiteboard for sketching. In addition, a low-reflective whiteboard was used to reduce light reflection. For applying the Douglas–Peucker algorithm for shape recognition, the threshold value for approximating the accuracy was set to 0.02. Figure 10 shows a visualization of the simulation results of the sketch-based environmental authoring tools.

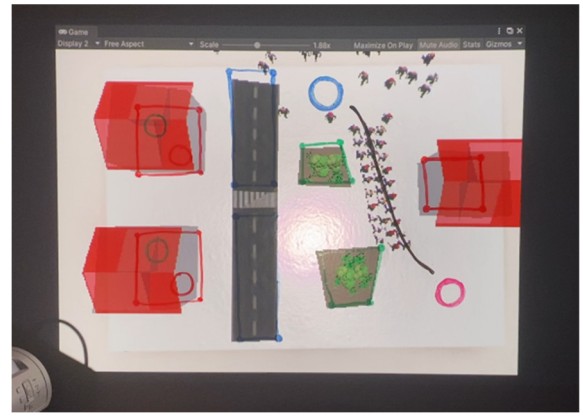
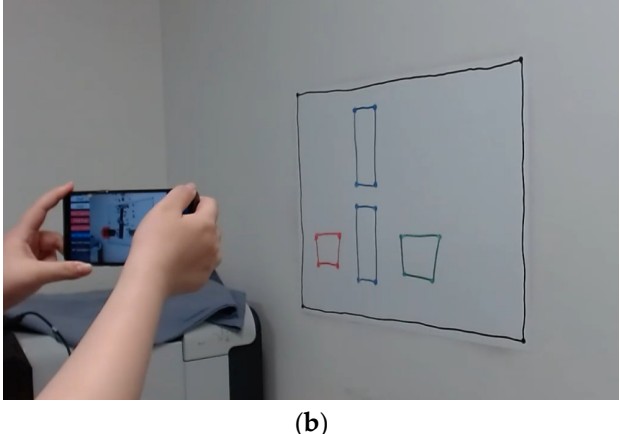

(**a**) (**b**)

**Figure 10.** Visualization of the simulation results as: (**a**) projector-based AR visualization; (**b**) portable device-based AR visualization. Note that AR can overlay the CG objects over the sketches that user drawn on the board.

Figure 11 shows a hand-drawn path and the crowd movement along the path. Note that inter-collisions between individual characters are prevented automatically.

### 4.2. Result of Experimentation

To verify our system, we provided users with a 2D screen-based and sketch-based environment authoring tools and conducted a comparative experiment to create an environment for a given situation. The main goal of the experiment was to compare the usability and learnability of sketch-based and 2D screen-based environment authoring tools. The results video (Video S1) can be viewed online (https://youtu.be/GdqMsQI_5MQ, accessed on 8 June 2022).

#### 4.2.1. Survey and Time Measurement Methods

After the users conducted the two experiments, they filled out a survey form. Twelve users participated in the survey, including two couples in their 20s, two couples in their 30s, and two pairs in their 50s. All participants have basic IT knowledge. They can handle a computer mouse and are familiar with the basic graphic user interface. The users worked

together in pairs to create a particular environment with two authoring tools. After the environment was created, they were able to visualize the entire environment and allow the virtual crowds react inside it.

In the first experiment, a cooperative experiment was conducted in pairs with the 2D screen-based environment authoring tool and the proposed sketch-based environment authoring tool. Users were asked to design both types of environments, referred to as A type and B type, with two different environment authoring tools. Figure 12 shows the two target environments.

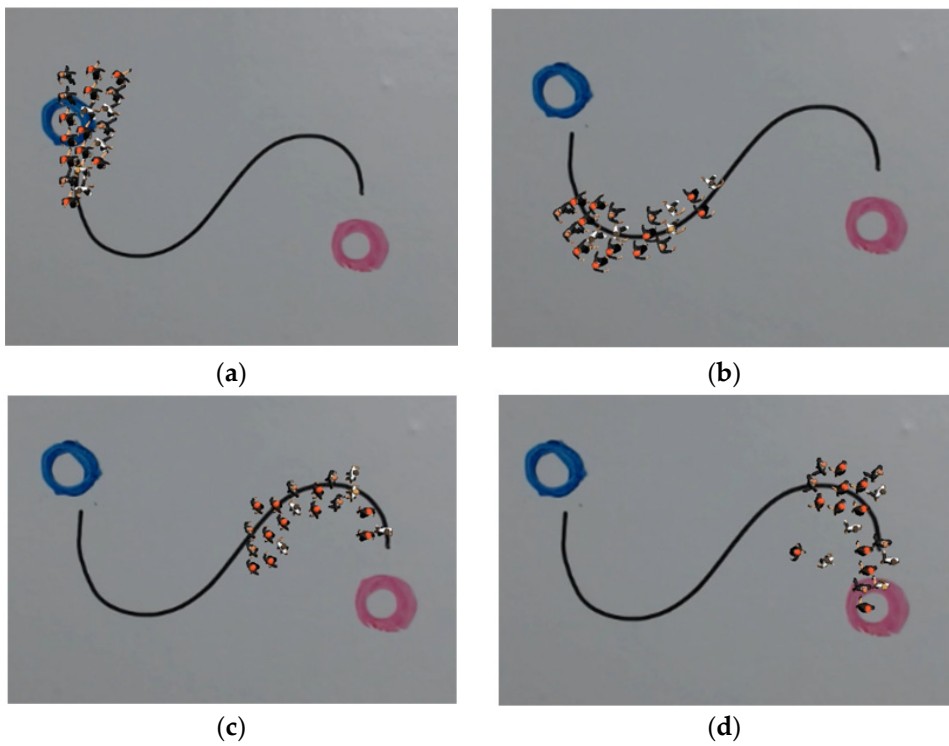

**Figure 11.** Crowds moving along the path from the starting and ending point. Note that there is no inter-penetration between characters during the animation. The figures from (**a**–**d**) are crowd simulation results over time where (**a**) is starting and (**d**) is the ending scene.

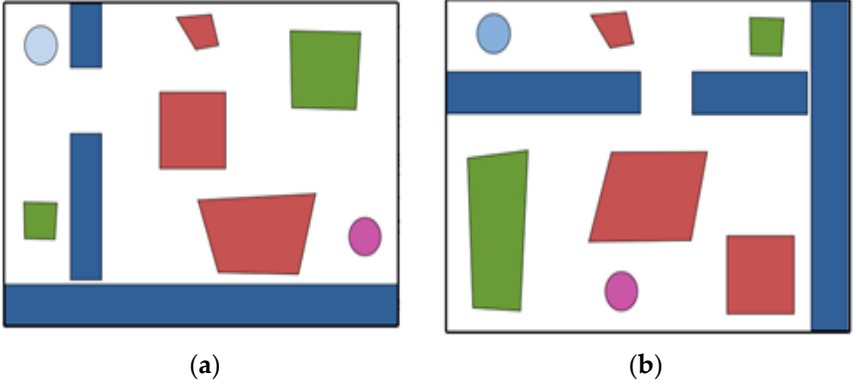

**Figure 12.** Two different target environments that users are required to design: (**a**) A Type; (**b**) B Type.

Users were instructed to select other tools when designing a second environment. For example, when a user designs environment A with a 2D screen-based authoring tool, they must design environment B with a sketch-based authoring tool, and vice versa. Before participating in the first experiment, sufficient explanation of two environmental creation activities was provided to the users. When users were asked to use the 2D screen-based

environment, GUI-based menus were shown on the screen, and mouse manipulation was used to specify all obstacles, starting points, and goal points. Subsequently, sketch-based environment creation was used to design the other environment on a whiteboard and a dry-erase marker. We measured the time required to complete the experiment.

In the second experiment, 12 users participated in the same experiments individually. Physical drawing was conducted until the desired environment was created without time measurement. After the environment design was completed, the users were able to observe the crowd movement in their environment using AR visualization.

### 4.2.2. Survey Result

Figure 13 and Table 3 present the results of the first cooperative experiment. Table 3 explains the four survey questionnaires. The users said that they required more explanation on how to create the environment when they used the 2D screen-based authoring tool before performing the experiment. In addition, they answered that their contributions to creating an environment using the sketch-based authoring tools were relatively higher than when they used the 2D screen-based tool. In light of this, they answered that creating an environment using sketches could be carried out in the future, rather than the 2D screen activities.

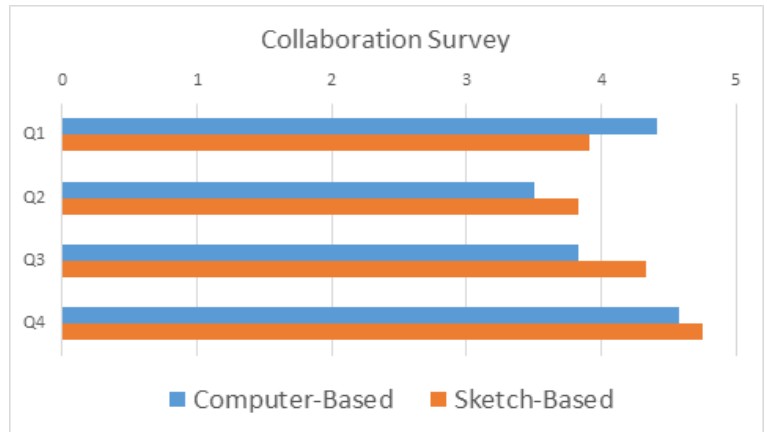

**Figure 13.** A result of survey about cooperative experiment.

**Table 3.** Survey contents of cooperative experiment.

| Question | Content |
| --- | --- |
| Q1 | Before the activity, it was necessary to understand the environmental creation method (building, tree, road arrangement, etc.). |
| Q2 | My contribution to environmental creation activities was high. |
| Q3 | I actively participated in the activity |
| Q4 | You will be able to do a series of processes by yourself in the future |

Table 4 lists the average time taken to complete the creation of the given environment with cooperation between participants. In the case of a 2D screen-based environment, only one mouse could be used because they had to share a single computer, and users who did not use the mouse provided location advice to the other user. When creating a sketch-based environment, participants can divide their roles and use separate dry-erase markers simultaneously. The sketch-based environment creation was completed approximately 1 min faster on average than 2D screen-based environment creation. Thus, it was confirmed that sketch-based environment creation can be performed faster than when using a 2D screen-based creation tool.

**Table 4.** Average time to create environment.

| Method | 2D Screen-Based | Sketch-Based |
| --- | --- | --- |
| Time (s) | 153 s | 64 s |

Figure 14 shows the results of a survey on learnability between the 2D screen and sketch-based environment creation methods. Learnability is a sub-characteristic of usability, representing how easy it is to learn to use the tools. Ten participants answered that the sketch-based environment was faster than the 2D screen-based environment. Thus, it was concluded that the sketch-based environment creation method was more efficient in terms of learnability.

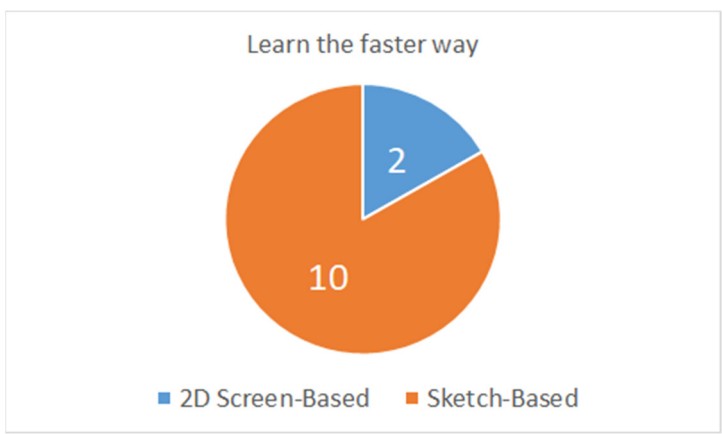

**Figure 14.** A result showing which tool was learned to be used faster.

Figure 15 shows the survey results after performing individual activities. Table 5 explains the survey questionnaires from Q1 to Q3 in Figure 14. After personal activities, users answered that they could install 3D objects at a more accurate location in the sketch-based crowd simulation than in the 2D screen-based crowd simulation. Users were often confused about how to specify each corner of the polygons of the objects using the mouse. Participants often discussed this by themselves. They answered that they required some assistance from experts when using the sketch-based tool. On the other hand, they appeared comfortable specifying polygons when they used sketch strokes.

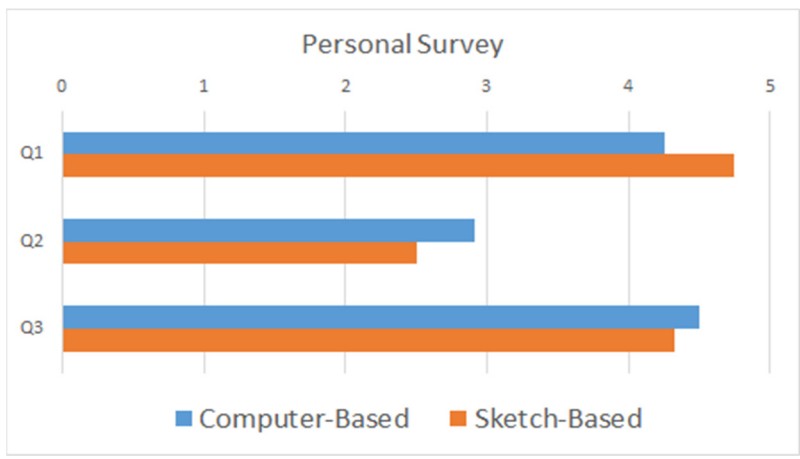

**Figure 15.** A result of survey about personal experiment.

**Table 5.** Survey contents of personal experiment.

| Question | Content |
| --- | --- |
| Q1 | It was possible to install sculptures (roads, trees, and buildings) that can be placed at the desired location. |
| Q2 | Next, an explanation from a technician is required to proceed with this activity. |
| Q3 | I was able to understand the finishing method of the program used in the activity well. |

The usability of the sketch-based environmental authoring tool was confirmed by Q1, as shown in Figure 15. Usability refers to the ease of use of tools. In Q1, the average sketch-based score was high. Users answered that the sketch-based environment authoring tool could place a 3D object at a more specific desired location than a 2D screen-based environment authoring tool. In the case of the pairs in their 50s, although GUI was simple (so as cannot affect to experiment) and they could handle the mouse, they complained of unfamiliarity in using the mouse interface and did not want to do so if they could. Thus, it was confirmed that sketch-based environmental authoring tools are easier to use than 2D screen-based tools.

The accessibility of the sketch-based environmental authoring tool can be confirmed by Q2 in Figure 15. Accessibility is used to check how easy it is to access the service without any inconvenience. With higher accessibility, the users will be more comfortable using it without restrictions. In Q2, users answered that if the 2D screen-based environmental creation authoring tool were to be used again, then technical assistance would be required. In the case of pairs that were unfamiliar with computers, when they use the basic 2D screen-based authoring tools, they answered that they still required some assistance from experts and repeated explanations would still be required the next time they used the tool. The need for a technician's assistance is a limiting factor when using the tool. From this, we can conclude that the sketch-based authoring tool has better accessibility than the 2D screen-based tool.

## 5. Conclusions and Future Work

In this study, the CoR-SketchAR system, a cooperative real-time sketch-based AR environment authoring tool, was proposed. To achieve this system, 3D buildings and paths through which crowds were followed were placed in the virtual environment by recognizing hand-drawn sketches and colors. Sketch-based environmental authoring tools using AR can be used in various fields. In this study, although an urban environment for a crowd was tested, it is possible to create environments of various places, such as coasts and mountains, through the recognition of other colors. It is also considered that users who are unfamiliar with the design would be able to gain experience in urban environment design and crowd simulation through relatively familiar sketches.

Through the qualitative experiments of cooperative activities between the computer-based and sketch-based environmental authoring tools, it was found that users who used the sketch-based authoring tools were more active than the computer-based ones. In addition, it was found that the sketch-based environment creation method was more accessible than the computer-based environment creation method based on the survey after the experiments. Finally, we found that the deployment time was significantly shorter than that of the 2D screen-based authoring tool.

Because the recognition performance changes depending on the color difference of the dry-erase markers and intensity of the light source, it is inconvenient to perform the color calibration step at run time. In this respect, a radiometer/photometer can provide more accurate color values. If more accurate color values can be derived, then the recognition performance would improve, regardless of the light conditions. One limitation of our approach is that our recognized shapes of the building with the current exit are simple but

actual buildings appear complicated. In the future, we would like to extend our research to produce more complex-shaped buildings by recognizing various types of additional shapes. By doing so, we believe it will be possible to simulate various situations, such as an evacuation from inside a complex building, through AR. Finally, in this study, only paths represented by black lines were supported to control the crowd. However, the crowd not only walks along the path but also performs various actions. For example, various activities occur in urban environments, such as waiting for someone in front of a building or stopping at a red light in front of a crosswalk. If various control tools can be created based on sketches using additional polygonal figure recognition, it will be possible to reproduce a considerably more realistic crowd.

**Supplementary Materials:** The following supporting information can be downloaded at: https://www.mdpi.com/article/10.3390/app12157416/s1, Video S1: CoR-SketchAR: Cooperative Sketch-based Realtime Augmented Reality Authoring Tool for Crowd Simulation.

**Author Contributions:** Conceptualization and methodology, G.K. and M.S.; software, G.K.; validation, G.K. and M.S.; formal analysis, G.K. and M.S.; investigation, G.K.; resources, G.K. and M.S.; data curation, G.K.; writing—original draft preparation, G.K. and M.S.; writing—review and editing, G.K. and M.S.; visualization, G.K.; supervision, M.S.; project administration, M.S. All authors have read and agreed to the published version of the manuscript.

**Funding:** This research was supported by a National Research Foundation of Korea (NRF) grant funded by the Korean government (MSIT) (No.2021R1A2C1012316).

**Institutional Review Board Statement:** Not applicable.

**Informed Consent Statement:** Informed consent was obtained from all subjects involved in the study.

**Data Availability Statement:** Not applicable.

**Conflicts of Interest:** The authors declare no conflict of interest.

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
