# Peer review of "CoR-SketchAR: Cooperative Sketch-Based Real-Time Augmented Reality Authoring Tool for Crowd Simulation"

_applsci, doi:10.3390/app12157416_

Round 1

Reviewer 1 Report

Well done and very interesting work. 

I suggest that you state the level of IT knowledge among the research participants. Can it influence the course of the experiment? Everyone can sketch, but working with a 2D graphic environment - not.

What is the justification for the selection of the research sample?

Author Response

Dear Reviwer

First, thank you for providing these insights.

We agree with you and have incorporated this suggestion through our paper.

We have responded specifically to each suggestion below, beginning with your own.

Reviwer’s suggestions:

  1. I suggest that you state the level of IT knowledge among the research participants. Can it influence the course of the experiment? Everyone can sketch, but working with a 2D graphic environment - not.

→ We added additional description of IT knowledge among the research participants on line 368-369 page 13. The participants have basic IT knowledge. All participants are familiar with the computer interface such as graphic user interface (GUI) and mouse interaction. We don’t think that their IT knowledge influence the result because they seemingly have equal prior knowledge.

What is the justification for the selection of the research sample?

→ If you saying why we selected those related work in Chapter 2, then we have selected those research samples based on the their goals and proposed technique that we adopted to apply. We have categorized all related work into four categories, which are real-time cooperative system, design in AR, crowd collision avoidance and path planning of crowds. Not only list up all related prior work, we compared their techniques with our method and analyzed the advantages and disadvantages.

Furthermore, we have made clear changes to the description in case of 50s which may seem ambiguous on two paragraph line 437-438 page 15. Additionally, we find four typos in page 6 Table 1. We change ‘upper’ to ‘under’.

Note that all highlighted sentences are modified in this revision process.

Reviewer 2 Report

a. Add the description of the method in the abstract.

b. Support the assertions in the introduction.

c. In the similar works section, add a conclusion about the findings.

d. Describe and justify the applied research method.

and. Some figures need to improve quality, they are not legible.

g. It is convenient to define objectives or hypotheses of the research and discuss the findings in the results section.

Author Response

Dear Reviewer

First, thank you for providing these insights.

We agree with you and have incorporated this suggestion through our paper.

We have responded specifically to each suggestion below, beginning with your own.

Reviewer’s suggestions:

  1. Add the description of the method in the abstract.

→ We added an additional description of the proposed method in the abstract line 18-24. We briefly explained the algorithms and how we validate it through experiments and user survey.

  1. Support the assertions in the introduction.

→ We added an assertion on line 79-81 saying that user-drawn sketch method is more convenient than the conventional methods and how the assertion may provide benefit to users on line 92-94.

  1. In the similar works section, add a conclusion about the findings.

→ We modified the whole related section. As a conclusion of each sub-section, we illustrated the drawbacks of the related work and compared those findings against our methods.

In section 2.1, we included the efficiency of collaborating works in line 107-108. We added the drawbacks of the related work in line 122-123. We also added a sentence about how our algorithm overcomes those disadvantages with the sketch-based method in line 123-125.

In section 2.2, we included the immersive advantage of AR system over other systems in line 147. We added the drawbacks of the related work in lines 148-149. We put additional sentences about why and how sketch-based methods are working in line 151-155.

In section 2.3, we included the disadvantage of the RVOE algorithm in line 172-173. We also included why we chose the RVO algorithm in our simulation in line 164-168.

In section 2.4, we included the disadvantage of related work in line 177-180 and 183-184, and then explained why and how we are use the curve sketch for crowd path in our system in line 176-177 and 185-187.

To be clear, the original section 2.5 is combined with the section 2.2 and 2.4. Note that the title of section 2.4 also changed too.

  1. Describe and justify the applied research method.
  2. Recognition the sketch shapes: we apply the Douglas-Peucker algorithm. When our algorithm recognizes the shape, it tries to detect the number of edges of the shape. To find contours of shape, the algorithm finds a set of continuous points from all detected points. Since the continuous points may include too many unnecessary points which are not part of the edge, we apply the Douglas-Peucker algorithm to produce a simplified edge that has fewer points than the original but still keeps the original’s characteristics/shape. Douglas-Peucker algorithm allows us to classify a linear line from a set of detected points by approximating the distance among vertices. we added a description in line 214-216.
  3. Circle shape recognition method: Unlike the rectangle, a circle is consisted of a lot of vertices. We apply the Hough gradient method to detect a circle. This method is based on the Hough transformation method but also included a module that measures the slope at the edge to determine if the point is relevant to the circle or not. We added description in line 217-221.
  4. RVO algorithm: Please refer to page 4 line 164-168. The RVO algorithm can get out of stuck situation earlier than other algorithms. That’s why we chose the algorithm. Although the RVOE algorithm is more similar to the real crowd but it has a disadvantage in terms of computation time. We mentioned it in line 170-174.
  5. Computing on HSV color space: Please refer to page 6 line 224-226. The HSV color space can separate the color and brightness value of each pixel. We used only the color value for recognizing the sketch to be less sensitive to the different light conditions.

.

And Some figures need to improve quality, they are not legible.

→ We increased the size of Figure 1,3,4,6,9,13,14 and 5 for better understanding, and replaced the Figure 2 with a new image. The Figure 2 means a conventional GUI interface on crowd simulation, which was asserted in 2.4 section.

  1. It is convenient to define objectives or hypotheses of the research and discuss the findings in the results section.

→ A hypotheses of research was put in line 195-198 beginning of section 3. We hypothesized that our proposed methods have advantages in terms of time cost and collaboration efficiency.

Additionally, we found four typos in page 6 in Table 1. We changed ‘upper’ to ‘under’.

Round 2

Reviewer 2 Report

The author followed the recommendations in this new version of the article. It has the quality to be published.